# Change in willingness for surgery and risk of joint replacement after an education and exercise program for hip/knee osteoarthritis: A longitudinal cohort study of 55,059 people

**Belinda J. Lawford**[1], **Ali Kiadaliri**[2], **Martin Englund**[2], **Kim L. Bennell**[1], **Rana S. Hinman**[1], **Michelle Hall**[3], **Andrea Dell'Isola**[2]*

**1** Centre for Health, Exercise and Sports Medicine, University of Melbourne, Melbourne Australia, **2** Clinical Epidemiology Unit, Orthopaedics, Department of Clinical Sciences Lund, Lund University, Lund, Sweden, **3** Sydney Musculoskeletal Health, The Kolling Institute, School of Health Sciences, University of Sydney, New South Wales, Australia

*andrea.dellisola@med.lu.se

## Abstract

### Background

Numerous studies report that education and exercise interventions can shift people's willingness to undergo joint replacement surgery for osteoarthritis. We aimed to investigate whether becoming unwilling to undergo surgery following an education and exercise intervention for hip and knee osteoarthritis is associated with lower probability of receiving actual surgery.

### Methods and findings

This was a register-based cohort study including people from the Swedish Osteoarthritis Register who underwent a 3-month education and exercise intervention for knee or hip osteoarthritis. Participants self-reported their willingness to have joint replacement surgery ('yes' or 'no') and were grouped based on their response pre- and post-intervention (always willing for surgery; became unwilling for surgery; never willing for surgery; became willing for surgery). Data on joint replacement surgery was obtained through the Swedish Arthroplasty Register. The probability and hazard of surgery occurring, as well as the mean time without surgery was calculated up to 5-years (primary outcome) and 9-years (secondary outcome) post-intervention. We adjusted for age, sex, body mass index (BMI), education, joint pain, quality of life, walking difficulties, number of prior visits with an orthopedic surgeon, prior joint surgeries in the knee or hip (other than joint replacement), and comorbidities. 55,059 people were included, 69% were female (N=37,739), with a mean age 66 years (standard deviation [SD] = 9.3), and a BMI of 27.5 (SD=4.9). In total, 70% (N=38,386) were never willing for surgery, 14% (N=7,736) were always willing for surgery, 10%

**Data availability statement:** The dataset of this includes data from the Swedish Osteoarthris Register, Swedish Drug Register, National Patient Register and Swedish Arthroplasty register, provided to the researchers through a restricted-access agreement that prevents sharing the dataset with a third party or publicly. Individual-level data of patients included in this paper after deidentification are considered sensitive and will not be shared. However, the individual-level data are accessible to authorized researchers after ethical approval and application to https://bestalladata.socialstyrelsen.se/bestalla-microdata-for-statistikandamal/ (contact: registerservice@socialstyrelsen.se) and https://etjanst.halsodatabestallning.vgregion.se/ (contact: regionalvardanalys@vgregion.se).

**Funding:** Greta and John Kock foundation, The Swedish Research Council (dnr: 2022–01507).

**Competing interests:** I have read the journal's policy and the authors of this manuscript have the following competing interests: ME is a consultant for Grunenthal Sweden AB and Key2Compliance.

(N = 5,649) became unwilling for surgery, and 6% (N = 3,288) became willing for surgery. Compared to those who were always willing for surgery, participants who became unwilling had a 20% (95% confidence interval [CI]: 18, 22%) lower probability of having surgery by 5-years post-intervention. This corresponded to delaying surgery by 1.1 (95% CI: 1.0, 1.1) years. Compared to those who were always willing for surgery, the hazard of surgery occurring at 1-year post-intervention was lower in those who became unwilling (hazard ratio (HR) 0.5 [95% CI: 0.4, 0.5]), though was then higher at 5-years (HR 1.4 [95% CI: 1.2, 1.7]). Estimates remained stable from 5 to 9 years. Limitations of our study include the inability to account for all potential confounders, and to infer the contribution of the intervention to change in willingness for surgery due to the absence of a control group. Data were collected in Sweden, generalisability to other countries may be limited.

## Conclusions

Becoming unwilling for joint replacement surgery following an education and exercise program for hip and knee osteoarthritis could reduce the number of joint replacement surgeries by 20% at 5 years post-intervention, with the possibility of maintaining most of this reduction up to 9 years post-intervention. Interventions that can shift willingness to undergo surgery may thus result in relevant delays and reductions in future joint replacements.

## Author summary

### Why Was This Study Done?

- Each year, millions of hip and knee joint replacement surgeries are performed for osteoarthritis worldwide, incurring substantial healthcare costs.

- First-line interventions like education and exercise interventions can shift self-reported willingness to undergo joint replacement surgery for osteoarthritis.

- However, it remains unclear whether change in self-reported willingness for surgery results in a reduction in the actual number of surgeries occurring in the short, medium, and long-term.

### What Did the Researchers Do and Find?

- This was a register-based cohort study including 55,059 people from Sweden who underwent a 3-month education and exercise intervention for knee or hip osteoarthritis.

- Participants self-reported their willingness to have joint replacement surgery ('yes' or 'no') and were grouped based on whether their willingness changed after completing the intervention.

PLOS Medicine

- People who became unwilling for surgery had a 20% lower probability of having actual surgery by 5-years after the intervention, compared to those who were always willing for surgery, corresponding to delaying the procedure by 1.8 years.

**What Do These Findings Mean?**

- Shifting willingness for joint replacement surgery in people with osteoarthritis could delay joint replacement surgery and lead to a reduction in the number of surgeries, potentially contributing to substantial economic savings.

- A simple question about willingness for surgery can be used as a proxy measure of progression to actual surgery in the short-medium term following an intervention, which could be used to help clinicians identify patients who may benefit from additional support to help them avoid or delay surgery in the future.

- We were unable to account for all potential confounders, and, due to the absence of a control group, we cannot infer whether the education and exercise intervention, or something else, contributed to a change in willingness for surgery.

## Introduction

Osteoarthritis (OA) of the knee and hip is one of the leading causes of pain and disability worldwide [1]. Globally, healthcare expenditure for OA is substantial, predominantly driven by the costs associated with total joint replacement surgery [2]. Each year, > 1.2 million hip and knee joint replacements are performed in the US alone, incurring $20 billion USD in healthcare costs [2,3]. In 2023, 17,089 hip joint replacements and 16,549 knee joint replacements for OA were performed in Sweden (10.5M inhabitants) [4], costing approximately $462 million USD [5]. With the ageing population and rising prevalence of obesity and sedentary lifestyles, rates of joint replacement surgery are projected to increase in the coming decades [6,7]. However, joint replacement surgery may not be effective for everyone [8], particularly for knee joint replacement where 1 in 4 patients report unsatisfactory symptom improvement [9–11]. Additional cost savings could therefore be achieved by providing non-surgical management of OA to prevent or delay joint replacement.

Clinical guidelines overwhelmingly recommend non-surgical, non-pharmacological treatment as core components of OA management [12–15], in particular education and advice, exercise, and weight loss for people who have overweight or obesity. Exercise is recommended for all people with hip and knee OA, irrespective of age, comorbidity, pain severity, or disability [12–15], due to its potential to improve joint pain and physical function [16,17]. There is also evidence that education and exercise programs can reduce or delay the need for joint replacement surgery among people with knee and hip OA [18–21].

Among the many factors influencing uptake of joint replacement surgery in people with knee and hip OA, self-reported willingness to have surgery has been shown to be the strongest predictor [22]. Numerous studies have suggested that exercise interventions can shift willingness to undergo surgery for knee and hip OA [23–27], with up to 71% of participants no longer desiring surgery after participation [21,23]. However, it remains unclear whether change in self-reported willingness for joint replacement surgery results in a reduction in the number of surgeries occurring in the short, medium, and long-term. Such information can be used to support the validity and usefulness of self-reported outcome measures related to willingness for surgery and guide future research on OA management. Thus, the aim of this study was to use long-term cohort data to investigate whether becoming unwilling to undergo surgery following an education and exercise intervention for hip and knee osteoarthritis is associated with lower probability of receiving actual surgery.

## Methods

### Study design

This is a longitudinal observational register-based cohort study using data from the Swedish Osteoarthritis Register (SOAR). The SOAR was started in 2008 and currently includes data from more than 120,000 individuals who sought treatment for OA in primary healthcare in Sweden [28]. To be eligible for inclusion in SOAR, participants were required to receive a clinical diagnosis of OA from primary or secondary care in Sweden and agree to participate in an education and exercise program. People with joint pain caused by another disease (e.g., hip fracture, inflammatory joint disease, cancer) were not eligible.

This study is reported as per the Reporting of Studies Conducted using Observational Routinely-Collected Data (RECORD) guideline [29] (S1 Appendix). Analyses were planned upon conception of this study. Data-driven changes to analyses were performed in response to peer reviewer comments, including adding another secondary analysis to stratify outcomes by pain severity.

### Education and exercise program

In Sweden, people with clinically confirmed OA can be referred by their healthcare provider to a publicly-funded education and exercise intervention, which is described in detail elsewhere [30,31]. Briefly, it comprises a mandatory education component (2 x 1-hour sessions) and a 3-month exercise component (up to 12 sessions with a physiotherapist) with the aim to improve participant ability to self-manage their OA. During education sessions, participants are provided with information about disease pathophysiology, the effectiveness and indication of OA treatments (including surgery, pharmacological management methods, and non-pharmacological management methods), benefits of exercise, self-management advice, and strategies around incorporating exercise into daily life. During the exercise component, participants receive a personalised program (based on individual needs, preferences, and level of physical function) to be completed three times per week, along with detailed exercise instructions. Participants provided data via an interview with a physiotherapist and self-reported questionnaire at the start ('baseline') and at the end ('post-intervention') of the program. At the first visit, the clinician and patient decided which was the most symptomatic joint to target in the intervention (knee or hip). In the case of OA affecting multiple joints, the most symptomatic joint was considered as the 'index' joint for the intervention. No major changes to this program were made during the timeframe of this study.

### Study sample

This project was approved by the Ethical Review Authority Board in Sweden (original application 2019–02570 and amendment 2020–04460). As this was a registry study, no additional consent was required from the participants for the specific research questions investigated. All participants had already agreed to allow their data to be used for research purposes at the time of their inclusion in the registry.

The sample used in this study consisted of people recorded in the SOAR between January 2008 and December 2018 who underwent the education and exercise program for their knee or hip OA (N = 71,089). We excluded anyone who had not been living in Sweden over the 10 years preceding the intervention (to minimize the chance that they had previously received a joint replacement, N = 511; Fig 1), anyone who had already had knee or hip joint replacement surgery (N = 4,996), and anyone who did not report their willingness for surgery at post-intervention (N = 2,570). Although follow-up is intended to occur at 3-months, some participants provided follow-up data earlier or later than that (i.e., they finished the intervention early or late due to scheduling issues or other disruptions) or did not provide follow-up data at all (i.e., had dropped out of the intervention). As such, to enhance data quality and maximise the generalizability of the results to similar interventions, we excluded participants who did not provide post-intervention within a reasonable timeframe (i.e., we excluded anyone who provided data >1 month before or more than 2 months after the 3-month follow-up time point; N = 7,944).

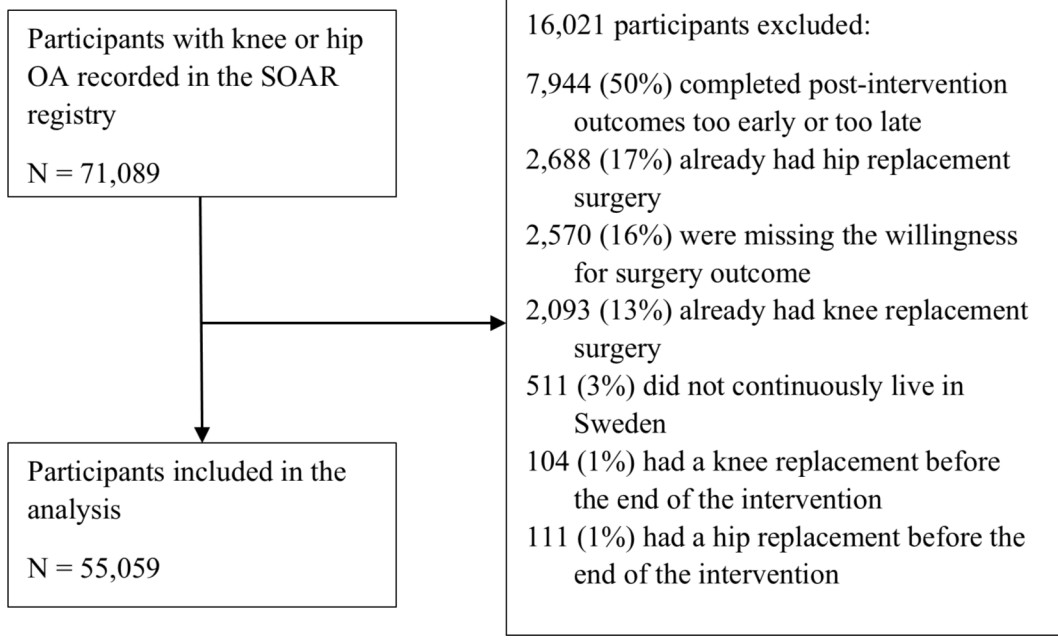

**Fig 1. Reasons for participant exclusion.** SOAR, Swedish Osteoarthritis Register.

## Exposure

Willingness to undergo joint replacement surgery was self-reported via questionnaire using the question "*Are your joint symptoms so severe that you wish to undergo surgery?*" ('Yes' or 'No') collected pre- and post-intervention. We grouped participants into four categories based on their self-reported willingness pre- and post-intervention: *YES-YES*, always willing for surgery; *NO-NO*, never willing for surgery; *YES-NO*, became unwilling for surgery, and; *NO-YES*, became willing for surgery.

## Outcome

Data relating to incident joint replacement surgery (i.e., the first joint replacement received by a person) for OA in either the hip or the knee was collected via the Swedish Arthroplasty Register. The register has a data complete-ness of 98%, covering nearly all the joint replacement surgeries performed in Sweden [4]. Our primary outcome was any joint replacement surgery due to OA up to five years post-intervention, as we assumed willingness for surgery was more likely to influence people's decision closer to the time point at which it was collected. Incident joint replacement surgery up to 9 years was reported in a secondary analysis (limited at 9 years as it was the last time point where at least 1% of the population had usable data; i.e., they did not experience the outcome and were not censored). All participants were followed from the 3-month post-intervention date until they either had joint replacement surgery for OA, death, joint replacement for reasons other than OA (e.g., fractures, cancer), relocation outside Sweden, or 31st December 2018, whichever came first. We did not match the index joint with the joint being replaced because surgery prioritisation in patients with multiple joint OA is not solely symptom-based; for example, a surgeon may choose to operate on the hip before the knee for biomechanical reasons, even if the knee is more severely affected. Moreover, the intervention focuses on overall symptoms and self-management, which are trans-ferable over multiple joints.

## Confounders

Based on prior evidence and direct acyclic graphs to identify confounders, we considered age, sex, body mass index (BMI), education (as proxy for socioeconomic status), joint pain (both at baseline and post-intervention to capture both absolute pain and change during the intervention; measured on 11-point Numeric Rating Scale ranging from 0 [no pain] to 10 [worst possible pain]), quality of life (both at baseline and post-intervention; measured on the Eq5D), walking difficulties (both at baseline and post-intervention; recorded as 'Yes' or 'No'), self-efficacy for pain (both at baseline and post-intervention; measured on Arthritis Self-Efficacy Scale), number of prior visits with an orthopedic surgeon in the year before the program, prior joint surgeries in the knee or hip (other than joint replacement), and comorbidities (Measured using Elixhauser score) as confounders [24,32–34]. All analyses were adjusted for confounders with the exception of self-efficacy for pain which was included as a confounder only in a secondary analysis. This was because the scale assessing self-efficacy for pain was no longer recorded from 2017 (i.e., all participants from 2017-2018 have the variable missing). Information about how each confounder variable was measured is included in S2 Appendix.

## Statistical analysis

We used flexible parametric survival models based on restricted cubic splines ("stpm2" command in Stata [35]) to estimate the association between change in willingness to undergo surgery during the intervention and the hazard of undergoing total joint replacement in the 5- and 9-years post-intervention.

Considering the low prevalence of missing exposure data (94% of the sample has complete data) and the large sample size, no imputation of missing data was performed. We tested the assumption of proportional hazard using Wald test for testing the statistical significance of an exposure*time interaction (i.e., the effect of the exposure on the outcome varied over time) and by assessing model fit. Model fit (for degrees of freedom 1−5 for the main model and up to *i-1* for the time varying factor where *i* is the degree of freedom of the main model) was assessed graphically by plotting the predicted cumulative hazard of models with and without time-varying factors against the Nelson-Aalen estimates and statistically by comparing the Bayesian information criterion (BIC) of the models with and without time varying coefficients (lower values indicate better fit). To choose the final model we used a parsimonious approach where the simplest model with the lowest BIC was selected [36].

The final model was adjusted for the confounders listed above. We then predicted standardised survival curves (using stpm2_standsurv command in Stata [37]), adjusted for all the listed confounders, under four counterfactual scenarios where all the participants are in the same willingness for surgery subgroup – i.e., never willing for surgery (Scenario 1), became willing for surgery (Scenario 2), became unwilling for surgery (Scenario 3) and always willing for surgery (Scenario 4). As we were interested in reducing the need for surgery, our main analysis compared the counterfactual scenario where everyone is willing for surgery at both pre- and post-intervention (always willing for surgery) with the scenario where people become unwilling for surgery. In a secondary analysis, we compared the counterfactual scenario where no one is willing for surgery at both pre- and post-intervention (never willing for surgery) with the scenario where people become willing for surgery. We contrasted these scenarios in terms of the following estimates:

(i)   proportion of individuals who have not had joint replacement surgery (i.e., probability of survival and difference in probability of survival between subgroups);

(ii)  hazard of receiving joint replacement surgery at a specific time point (i.e., instantaneous hazard of surgery per 1,000 people among those who have not yet had surgery at that time, and hazard ratios between subgroups), and;

(iii) mean time without having joint replacement surgery (i.e., restricted mean survival time and difference in mean survival time between subgroups).

We also conducted additional secondary analyses: (**i**) repeated the analysis extending the follow-up time up to 9 years; (**ii**) repeated the analysis with follow-up to 9 years stratified by the index joint (most symptomatic joint, knee or hip); (**iii**) repeated the analysis with follow-up to 9 years stratified by pain severity at baseline (pain above median considered severe and below median as mild), and; (**iv**) adjusted the main analysis for self-efficacy at baseline and post-intervention.

## Results

In total, 55,059 individuals with hip (N = 17,216; 31%) or knee (N = 37,843; 69%) OA from the SOAR were included in this study (S3 Appendix). Most participants were female (N = 37,739; 69%), with a mean age 66.1 years (standard deviation [SD] = 9.3), and a BMI of 27.5 (SD = 4.9). At baseline, mean (SD) joint pain was 5.3 out of 10 (SD = 2.0) and reduced to 4.3 (SD = 2.3) at 3-month post-intervention.

More than two-thirds of participants (N = 38,386, 70%; Table 1) were never willing for surgery, 14% (N = 7,736) were always willing, 10% (N = 5,649) became unwilling, and 6% (N = 3,288) became willing. Compared to those who were always willing for surgery, those who became unwilling showed milder symptoms post-intervention and were more likely to have knee OA. The opposite was observed when comparing those who were never willing for surgery to those who became willing, in that those who became willing had more severe symptoms and were less likely to have knee OA.

### Primary analysis: Comparing those who were always willing for surgery to those who became unwilling, up to 5 years post-intervention

The cumulative number of surgeries for each year post-intervention within each subgroup is presented in S4 Appendix. Among those who became unwilling for surgery, the probability of having undergone surgery by 1- and 5-years post-intervention was 23% (95% CI: 22, 24%) and 20% (95% CI: 18, 22%) lower, respectively, than those who were always willing for surgery (Table 2 and Fig 2). For those who became unwilling for surgery, the hazard of having surgery at 1-year post-intervention was half that of those who were always willing for surgery (hazard ratio 0.5 [95% CI: 0.4, 0.5]; Table 3 and Fig 2). At 3-years post-intervention, the hazard of surgery occurring was similar in both subgroups (1.1 [95% CI: 1.0, 1.2]), and, at 5-years, was higher among those who became unwilling for surgery (1.4 [95% CI: 1.2, 1.7]). Over 5-years, becoming unwilling for surgery was associated with surgery occurring 1.1 (95% CI: 1.0, 1.1) years later than those who were always willing for surgery (Table 4 and Fig 2).

### Secondary analysis: Comparing those who were never willing for surgery to those who become willing, up to 5-years post-intervention

Among those who became willing for surgery, the probability of having undergone surgery at 5-years post-intervention was 23% (95% CI: 21, 25%) higher than those who were never willing for surgery (Table 2 and Fig 2). This corresponded to surgery occurring an average of 1.1 (95% CI: 1.0, 1.1) years earlier (Table 4 and Fig 2).

### Secondary analysis: Long-term outcomes up to 9-years post-intervention

Among those who became unwilling for surgery, the probability of having undergone surgery at 9-years post-intervention was 15% (95% CI: 13, 18%) lower than those who were always willing for surgery (Table 2). This corresponded to surgery occurring an average of 1.8 (95% CI: 1.6, 1.9) years later (Table 4). The opposite trend was observed when comparing those who became willing for surgery to those who were never willing, who had a 19% (95% CI: 17, 21%) higher probability of having undergone surgery (Table 2), corresponding to surgery occurring an average of 1.9 (95% CI: 1.8, 2.1) years earlier (Table 4).

### Secondary analysis: Differences between hip and knee OA

Overall trends were similar between those with knee and hip OA. However, compared to people with knee OA, the magnitude of differences between subgroups in terms of probability and hazard of surgery occurring, as well as amount of time

**Table 1.** Demographics and sample characteristics (N = 55,059).

| | Never willing for surgery | Became willing for surgery | Became unwilling for surgery | Always willing for surgery | Total |
|---|---|---|---|---|---|
| | N: 38,386 | N: 3,288 | N: 5,649 | N: 7,736 | N: 55,059 |
| **Sex, N (%)** | | | | | |
| Male | 10,483 (27) | 1,153 (35) | 2,285 (40) | 3,399 (44) | 17,320 (32) |
| Female | 27,903 (73) | 2,135 (65) | 3,364 (60) | 4,337 (56) | 37,739 (69) |
| Age (years), mean (SD) | 66.2 (9.3) | 66.8 (9.2) | 65.0 (9.5) | 65.9 (9.5) | 66.1 (9.3) |
| Body mass index, mean (SD) | 27.2 (4.8) | 27.9 (4.9) | 28.2 (4.9) | 28.5 (5.1) | 27.5 (4.9) |
| **Education attainment, N (%)** | | | | | |
| 0–9 years | 11,914 (31) | 1,156 (35) | 1,982 (35) | 2,914 (38) | 17,966 (33) |
| 10–14 years | 14,387 (38) | 1,258 (39) | 2,263 (40) | 3,176 (41) | 21,084 (38) |
| >14 years | 11,952 (31) | 852 (26) | 1,388 (25) | 1,619 (21) | 15,811 (29) |
| **Affected joint, N (%)** | | | | | |
| Hip | 11,132 (29) | 1,345 (41) | 1,566 (28) | 3,173 (41) | 17,216 (31) |
| Knee | 27,254 (71) | 1,943 (59) | 4,083 (72) | 4,563 (59) | 37,843 (69) |
| **Walking difficulties, N (%)** | | | | | |
| No | 10,016 (26) | 346 (11) | 409 (7) | 248 (3) | 11,019 (20) |
| Yes | 28,089 (74) | 2,928 (89) | 5,206 (93) | 7,460 (97) | 43,683 (80) |
| Pain at baseline[*], mean (SD) | 4.9 (2.0) | 5.8 (1.8) | 6.3 (1.7) | 6.8 (1.6) | 5.3 (2.0) |
| Pain at follow-up[*], mean (SD) | 3.7 (2.0) | 6.3 (1.8) | 4.2 (2.0) | 6.5 (1.8) | 4.3 (2.3) |
| Quality of life at baseline[£], mean (SD) | 0.7 (0.2) | 0.6 (0.2) | 0.5 (0.2) | 0.4 (0.3) | 0.6 (0.2) |
| Quality of life at follow-up[£], mean (SD) | 0.7 (0.1) | 0.5 (0.2) | 0.7 (0.2) | 0.5 (0.3) | 0.7 (0.2) |
| Pain self-efficacy[§], mean (SD) | 67.2 (17.3) | 61.3 (17.3) | 55.6 (18.0) | 49.6 (18.4) | 63.3 (18.7) |
| **Number of comorbidities[¥], N (%)** | | | | | |
| 0 | 25,565 (67) | 1,968 (60) | 3,512 (62) | 4,666 (60) | 35,711 (65) |
| 1 | 7,068 (18) | 677 (21) | 1,088 (19) | 1,528 (20) | 10,361 (19) |
| 2 | 3,304 (9) | 355 (11) | 587 (10) | 781 (10) | 5,027 (9) |
| 3 + | 2,449 (6) | 288 (8) | 462 (9) | 761 (10) | 3,960 (7) |
| Comorbidity Elixhauser[¥] score (0–31), mean (SD) | 0.6 (1.0) | 0.8 (1.2) | 0.7 (1.2) | 0.8 (1.2) | 0.6 (1.1) |
| **Visited surgeon previous year, N (%)** | | | | | |
| No | 35,137 (92) | 2,814 (86) | 4,777 (85) | 6,199 (80) | 48,927 (89) |
| Yes | 3,249 (8) | 474 (14) | 872 (15) | 1,537 (20) | 6,132 (11) |
| Number of orthopaedic surgeon visits during the year prior to the intervention, mean (SD) | 0.1 (0.4) | 0.2 (0.5) | 0.2 (0.5) | 0.3 (0.6) | 0.1 (0.4) |

SD: Standard Deviation

[*]Measured on 11-point Numeric Rating Scale ranging from 0 (no pain) to 10 (worst possible pain)

[£]Measured on the Eq5D; scores range 0-1.0 (higher values represent better quality of life)

[§]Measured on Arthritis Self-Efficacy Scale; scores range 10–100 (higher values represent better self-efficacy)

[¥]Measured using Elixhauser score, ranging 0–31 [38]

without surgery, was greater in those with hip OA (e.g., among those with hip OA, those who became unwilling for surgery had a 23% [95% CI: 19, 26%] lower probability of having surgery at 5 years than those who were always willing, compared to 17% [95% CI: 14, 19%] among those with knee OA; S5-S8 Appendices).

**Table 2. Adjusted\* proportion of participants who had not had surgery following the intervention.**

| Years post-intervention | Proportion, % (95% CI) | Proportion, % (95% CI) | Proportion difference, % (95% CI) | Proportion, % (95% CI) | Proportion, % (95% CI) | Proportion difference, % (95% CI) |
|---|---|---|---|---|---|---|
| | Always willing for surgery | Became unwilling for surgery | Became unwilling VS always willing | Never willing for surgery | Became willing for surgery | Became willing VS Never willing |
| 1 | 71 (70, 72) | 94 (94, 95) | 23 (22, 24) | 98 (97, 98) | 78 (77, 79) | −20 (−21, −18) |
| 2 | 62 (60, 63) | 86 (85, 87) | 24 (23, 26) | 93 (93, 93) | 69 (67, 70) | −24 (−26, −23) |
| 3 | 57 (56, 58) | 80 (79, 81) | 23 (21, 25) | 89 (89, 90) | 64 (63, 66) | −25 (−26, −23) |
| 4 | 55 (53, 56) | 76 (75, 77) | 22 (20, 23) | 86 (86, 87) | 62 (60, 64) | −24 (−26, −22) |
| 5 | 53 (51, 54) | 73 (72, 74) | 20 (18, 22) | 84 (83, 84) | 60 (59, 62) | −23 (−25, −21) |
| 6 | 52 (50, 53) | 70 (69, 72) | 19 (17, 21) | 82 (81, 82) | 59 (58, 61) | −22 (−24, −20) |
| 7 | 51 (49, 52) | 68 (66, 70) | 18 (15, 20) | 80 (79, 80) | 59 (57, 61) | −21 (−23, −19) |
| 8 | 50 (48, 51) | 66 (64, 68) | 16 (14, 19) | 78 (77, 79) | 58 (56, 60) | −20 (−22, −18) |
| 9 | 49 (47, 51) | 64 (62, 66) | 15 (13, 18) | 76 (75, 77) | 57 (55, 59) | −19 (−21, −17) |

CI: confidence interval

\*Adjusted by: age, sex, body mass index (BMI), education, joint pain (both at baseline and post-intervention), quality of life (both at baseline and post-intervention) walking difficulties (both at baseline and post-intervention), number of prior visits with an orthopedic surgeon in the year before the intervention, prior joint surgeries in the knee or hip (other than joint replacement), and comorbidities.

'Always willing for surgery' = indicated they were willing for surgery both before and after the intervention. 'Became unwilling for surgery' = indicated they were willing for surgery before the intervention, but unwilling after. 'Never willing for surgery' = indicated they were unwilling for surgery both before and after the intervention. 'Became willing for surgery' = indicated they were unwilling before the intervention, but willing after.

## Secondary analysis: Differences between those with severe and mild pain at baseline

Overall trends were similar to those observed in the main analysis. However, people with more severe pain had a higher probability of undergoing surgery than those with milder pain, resulting in a larger difference in probability when becoming unwilling (among those with severe pain, those who became unwilling for surgery had a 26% [95% CI: 23, 29%] lower probability of having surgery at 5 years than those who were always willing, compared to 16% [95% CI: 13, 18%] among those with mild pain; S9 Appendix).

## Secondary analysis: Adjustment for self-efficacy

Results did not change when adjusting for arthritis self-efficacy, except for a decrease in precision due to the reduced sample size (S10 Appendix).

## Discussion

The aim of this study was to investigate whether becoming unwilling to undergo surgery following a 3-month education and exercise intervention for hip and knee osteoarthritis is associated with lower probability of receiving actual surgery. We found that becoming or remaining unwilling to have joint replacement surgery could delay the procedure by more than 1 year and lead to a 20% reduction in surgeries 5 years after the intervention; with the possibility that this is maintained up to 9 years after the intervention.

We found that, for all subgroups, the hazard of surgery (for anyone who had not already had surgery at that time point) peaked around 1-year post-intervention and then declined, becoming similar between all subgroups by 3-years post-intervention. This reflects prior work [39,40,41] which also found that the rate of progression to surgery declines around 3-years after a conservative management program for people with hip and knee OA. This may be because most people who are eligible and willing for surgery tend to have it early (within the first 3 years after an intervention, particularly in publicly funded healthcare systems) [39,40,41]. This also helps explains why the hazard of surgery at 5-years

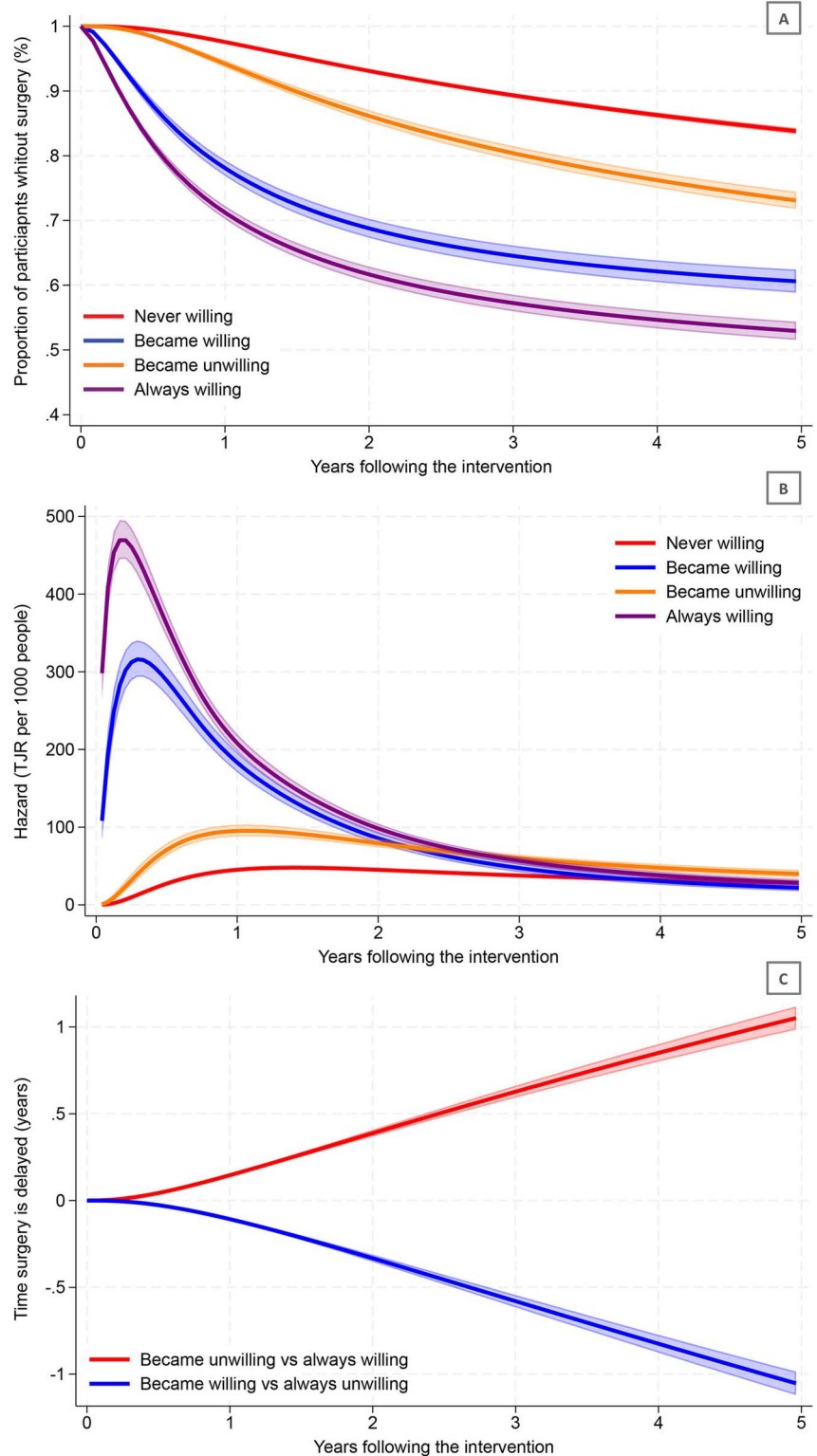

**Fig 2. Adjusted* (A) Proportion of participants who had not had surgery; (B) hazard^ of having surgery, and; (C) difference in amount of time surgery is delayed# following the intervention.** Lines represent point estimates, shaded areas represent the 95% Confidence Intervals of the estimates. TJR, total joint replacement. Lines represent point estimates, shaded areas represent the 95% Confidence Intervals of the estimates. TJR = total

joint replacement. *Adjusted by: age, sex, body mass index (BMI), education, joint pain (both at baseline and post-intervention), quality of life (both at baseline and post-intervention) walking difficulties (both at baseline and post-intervention), number of prior visits with an orthopedic surgeon in the year before the intervention, prior joint surgeries in the knee or hip (other than joint replacement), and comorbidities. ^Number of joint replacement surgeries per 1000 people among those who had not already had surgery at that time-point). 'Always willing for surgery' = indicated they were willing for surgery both before and after the intervention. 'Became unwilling for surgery' = indicated they were willing for surgery before the intervention, but unwilling after. 'Never willing for surgery' = indicated they were unwilling for surgery both before and after the intervention. 'Became willing for surgery' = indicated they were unwilling before the intervention, but willing after*.

Table 3. Adjusted* hazard of having surgery^ following the intervention.

| Years post-intervention | Hazard (95%CI) | Hazard (95%CI) | Hazard ratio (95%CI) | Hazard (95%CI) | Hazard (95%CI) | Hazard ratio (95%CI) |
|---|---|---|---|---|---|---|
| | Always willing for surgery | Became unwilling for surgery | Became unwilling VS always willing | Never willing for surgery | Became willing for surgery | Became willing VS never willing |
| 1 | 206 (195, 218) | 95 (88, 103) | 0.5 (0.4, 0.5) | 45 (43, 47) | 182 (170, 195) | 4.0 (3.7, 4.4) |
| 2 | 97 (91, 103) | 79 (74, 84) | 0.8 (0.7, 0.9) | 45 (43, 47) | 85 (78, 92) | 1.9 (1.7, 2.1) |
| 3 | 56 (51, 61) | 60 (55, 64) | 1.1 (1.0, 1.2) | 38 (36, 39) | 47 (41, 53) | 1.2 (1.1, 1.4) |
| 4 | 38 (33, 42) | 47 (43, 52) | 1.3 (1.1, 1.5) | 32 (30, 33) | 30 (25, 36) | 0.9 (0.8, 1.1) |
| 5 | 28 (24, 33) | 39 (35, 45) | 1.4 (1.2, 1.7) | 28 (26, 30) | 22 (17, 27) | 0.8 (0.6, 1.0) |
| 6 | 23 (19, 27) | 35 (30, 40) | 1.5 (1.2, 1.9) | 25 (24, 27) | 17 (13, 22) | 0.7 (0.5, 0.9) |
| 7 | 19 (16, 23) | 31 (27, 37) | 1.6 (1.3, 2.1) | 24 (22, 26) | 14 (11, 19) | 0.6 (0.4, 0.8) |
| 8 | 17 (14, 20) | 29 (24, 34) | 1.7 (1.3, 2.2) | 22 (20, 24) | 12 (9, 16) | 0.5 (0.4, 0.7) |
| 9 | 15 (12, 19) | 27 (23, 32) | 1.8 (1.4, 2.3) | 21 (19, 23) | 11 (8, 15) | 0.5 (0.4, 0.7) |

CI: confidence interval

*Adjusted by: age, sex, body mass index (BMI), education, joint pain (both at baseline and post-intervention), quality of life (both at baseline and post-intervention) walking difficulties (both at baseline and post-intervention), number of prior visits with an orthopedic surgeon in the year before the intervention, prior joint surgeries in the knee or hip (other than joint replacement), and comorbidities.

^Number of joint replacement surgeries per 1,000 people among those who had not already had surgery at that time point)

'Always willing for surgery' = indicated they were willing for surgery both before and after the intervention. 'Became unwilling for surgery' = indicated they were willing for surgery before the intervention, but unwilling after. 'Never willing for surgery' = indicated they were unwilling for surgery both before and after the intervention. 'Became willing for surgery' = indicated they were unwilling before the intervention, but willing after.

post-intervention were higher in those who became unwilling for surgery than those who were always willing for surgery (i.e., a larger proportion of those who were always willing had already undergone surgery in the first 3-years, and had not 'survived' to 5-years post-intervention). In the years after the intervention, symptom progression as well as underutilization of self-management strategies (which may contribute to diminishing effects on symptoms) [42,43] may have also contributed to reconsideration about willingness for surgery. To minimise risk of surgery reconsideration in the months and years following an intervention, longer interventions, additional contact with care providers, booster sessions, or re-completion of the education and exercise intervention might be necessary [27,44].

Our findings suggest that a simple question about willingness for surgery can be used as a proxy measure of progression to surgery in the short-medium term following an intervention. As such, researchers and clinicians should consider using such an outcome to evaluate the effectiveness of OA management interventions. This may also help clinicians identify those who remain willing for surgery post-intervention, and may therefore benefit from additional interventions to help them avoid or delay surgery in the future. However, there is some variability in the way in which participant willingness for surgery has been measured. In our study, participants were asked "*Are your joint symptoms so severe that you wish to undergo surgery?*", responding either 'Yes' or 'No', which is similar to some prior work [45]. Other studies have used slightly different questions and/or outcome scales (e.g., 5-point Likert scales) [23,24]. The best way in which to measure willingness to undergo surgery, and detect changes in willingness, needs further investigation.

**Table 4. Adjusted\* amount of time without surgery (mean survival time) and average time surgery can be delayed (difference in mean survival time) following the intervention.**

| Years post-intervention | Mean survival time (years) | Mean survival time (years) | Difference in mean survival time (years) | Mean survival time (years) | Mean survival time (years) | Difference in mean survival time (years) |
|---|---|---|---|---|---|---|
| | Always willing for surgery | Became unwilling for surgery | Became unwilling VS always willing | Never willing for surgery | Became willing for surgery | Became willing VS never willing |
| 1 | 0.8 (0.8, 0.9) | 1.0 (1.0, 1.0) | 0.2 (0.1, 0.2) | 1.0 (1.0, 1.0) | 0.9 (0.9, 0.9) | −0.1 (−0.1, −0.1) |
| 2 | 1.5 (1.5,1.5) | 1.9 (1.9, 1.9) | 0.4 (0.4, 0.4) | 2.0 (2.0, 2.0) | 1.6 (1.6, 1.7) | −0.3 (−0.4, −0.3) |
| 3 | 2.1 (2.1, 2.1) | 2.7 (2.7, 2.8) | 0.6 (0.6, 0.7) | 2.9 (2.9, 2.9) | 2.3 (2.3, 2.3) | −0.6 (−0.6, −0.6) |
| 4 | 2.7 (2.6, 2.7) | 3.5 (3.5, 3.6) | 0.9 (0.8, 0.9) | 3.8 (3.8, 3.8) | 2.9 (2.9, 3.0) | −0.8 (−0.9, −0.8) |
| 5 | 3.2 (3.2, 3.3) | 4.3 (4.2, 4.3) | 1.1 (1.0, 1.1) | 4.6 (4.6, 4.6) | 3.6 (3.5, 3.6) | −1.1 (−1.1, −1.0) |
| 6 | 3.7 (3.6, 3.8) | 4.9 (4.9, 5.0) | 1.3 (1.2, 1.3) | 5.4 (5.4, 5.4) | 4.1 (4.0, 4.2) | −1.3 (−1.4, −1.2) |
| 7 | 4.2 (4.1, 4.3) | 5.6 (5.6, 5.7) | 1.4 (1.3, 1.5) | 6.2 (6.3, 6.2) | 4.7 (4.6, 4.8) | −1.5 (−1.6, −1.4) |
| 8 | 4.7 (4.6, 4.8) | 6.3 (6.2, 6.4) | 1.6 (1.5, 1.7) | 7.0 (7.0, 7.0) | 5.3 (5.2, 5.4) | −1.7 (−1.8, −1.6) |
| 9 | 5.2 (5.1, 5.3) | 7.0 (6.9, 7.1) | 1.8 (1.6, 1.9) | 7.8 (7.7, 7.8) | 5.9 (5.7, 6.0) | −1.9 (−2.2, −1.8) |

\*Adjusted by: age, sex, body mass index (BMI), education, joint pain (both at baseline and post-intervention), quality of life (both at baseline and post-intervention) walking difficulties (both at baseline and post-intervention), number of prior visits with an orthopedic surgeon in the year before the intervention, prior joint surgeries in the knee or hip (other than joint replacement), and comorbidities.

'Always willing for surgery' = indicated they were willing for surgery both before and after the intervention. 'Became unwilling for surgery' = indicated they were willing for surgery before the intervention, but unwilling after. 'Never willing for surgery' = indicated they were unwilling for surgery both before and after the intervention. 'Became willing for surgery' = indicated they were unwilling before the intervention, but willing after.

Our work has implications for clinical practice. Our findings suggest that shifting willingness for joint replacement surgery in people eligible for an education and exercise intervention could delay joint replacement surgery and lead to fewer surgeries (>20% reduction over 5 years). This could potentially contribute to substantial economic savings. There are numerous factors that influence change in willingness for surgery. Multiple studies report that education, exercise, and/or weight loss interventions can reduce willingness for surgery [23–27]. People who experience improvement in symptoms (e.g., self-reported pain and physical function, and arthritis self-efficacy) [24,46,47], do not have walking difficulties [48], are younger [24], have lower pain at baseline [24,47], and complete the entire treatment program (i.e., attend the final appointment) [24] are more likely to be, or become, unwilling for surgery. Given that our estimates are adjusted for change in symptoms, the actual impact of an intervention aimed at changing willingness for surgery may result in an even larger reduction in actual surgeries, as many participants are likely to experience improvements in symptoms after the intervention. Our secondary analysis also showed that targeting those with severe pain at baseline may result in an even greater reduction in surgeries in the future. However, clinicians should be aware that people with more severe pain still have a higher probability of undergoing surgery than those with milder pain (as shown by our secondary analysis) and therefore may need additional support to manage their condition, maintain quality of life, and avoid low-value care [49], even if their willingness for surgery changes. Other factors, such as having peers or family members who have had a joint replacement, interaction with a surgeon, and receiving a recommendation for surgery, may also play a role in willingness for surgery. It is also important to acknowledge that changing willingness for surgery can be challenging – only 42% of participants in our cohort who were originally willing for surgery at baseline became unwilling at post-intervention.

Our findings have implications for future research. Further work is needed to evaluate the potential mechanisms by which education and exercise interventions can contribute to changes in willingness for surgery, as well as the characteristics of those who do and do not change their willingness. Future research should also evaluate new cost-effective ways of further shifting surgery willingness, such as through use of tools like decision aids [50] or predictive tools (which provide personalised information about the likelihood in improvement after surgery, based on the participant's age, sex,

and baseline symptoms) [45].Finally, future research should use qualitative methods to explore the perceptions and experiences of those that became unwilling for surgery, including why their beliefs changed and what support is important to maintain this change in willingness in the long-term.

Our study has strengths and limitations. One strength of our approach is the use of a large sample of real-world data. However, some unknown and unobserved confounders – such as social factors, risk-taking behaviours, duration of symptoms, total number of joints affected – are likely to be present and thus causality cannot be determined. Furthermore, due to the absence of a control group, we cannot infer whether the intervention, or something else (e.g., undergoing other treatments during the study period), contributed to the shifts in willingness to undergo surgery. We did not account for the variability introduced by different hospitals, where varying protocols, resources, and local hospital culture can influence the probability of a patient receiving surgery. Ignoring these clustering effects may lead to a biased association between willingness for surgery status and the probability of receiving actual surgery. Moreover, we did not account for variability in attendance at the education and exercise program. However, prior work using data from the SOAR has suggested that attendance is only minimally associated with symptoms [51] and outcomes [52]. While it is highly likely that the self-reported willingness to undergo surgery impacted the hazard of receiving the operation in the short term (1–2 years), caution is needed when inferring causality for longer periods. We excluded participants who did not provide data at post-intervention, however missing data accounted for less than 5% of the total sample and is therefore unlikely to have created significant bias. As our data were collected in a clinical setting, there was some variability in the precise time point at which follow-up data was collected. We did not distinguish whether participants received joint replacement surgery to their index joint or to another affected joint (e.g., a person with both hip and knee OA might have nominated their knee as their index joint, but later undergone joint replacement surgery for their hip). However, we do not believe this introduces bias as, of those who underwent joint replacement surgery, 97% of individuals with knee OA and 90% of those with hip OA received the joint replacement on their index joint. Finally, our data were collected in a Swedish healthcare setting and in people who were eligible for an education and exercise intervention, where that intervention is publicly funded and requires minimal to no out-of-pocket costs to participate. As such our results may not be generalisable to other countries with different healthcare systems and cultures.

In conclusion, we found that becoming unwilling for joint replacement surgery following an education and exercise program for hip and knee osteoarthritis could reduce the number of joint replacement surgeries by 20% at 5 years post-intervention, with the possibility of maintaining most of this reduction up to 9 years post-intervention.

## Ethics committee approval

This project was approved by the Ethical Review Authority Board in Sweden (original application 2019–02570 and amendment 2020–04460). As this was a registry study, no additional consent was required from the participants for the specific research questions investigated. All participants had already agreed to allow their data to be used for research purposes at the time of their inclusion in the registry.

## Supporting information

**S1 Appendix. RECORD statement.**
(PDF)

**S2 Appendix. List of confounders included in the analyses.**
(PDF)

**S3 Appendix. Demographics and sample characteristics.**
(PDF)

**S4 Appendix. Cumulative number of joint replacement surgeries from 1 to 9 years after the intervention (N = 55,059).**
(PDF)

**S5 Appendix. Adjusted differences in proportion of participants who had not had surgery following the intervention, categorized by knee and hip osteoarthritis.**
(PDF)

**S6 Appendix. Adjusted differences in hazard of having surgery following the intervention, categorised by knee and hip osteoarthritis.**
(PDF)

**S7 Appendix. Adjusted difference in average time surgery can be delayed following the intervention, categorised by knee and hip osteoarthritis.**
(PDF)

**S8 Appendix. Adjusted (A) proportion of participants who had not had surgery; (B) hazard of having surgery, and; (C) average time surgery can be delayed following the intervention, categorized by knee and hip osteoarthritis.**
(PDF)

**S9 Appendix. Adjusted proportion of participants with severe and mild pain who had not had surgery following the intervention.**
(PDF)

**S10 Appendix. Adjusted (A) proportion of participants who had not had surgery, and; (B) hazard of having surgery following the intervention, when adjusting for self-efficacy.**
(PDF)

## Author contributions

**Conceptualization:** Belinda J Lawford, Ali Kiadaliri, Andrea Dell'Isola.

**Formal analysis:** Ali Kiadaliri, Andrea Dell'Isola.

**Funding acquisition:** Martin Englund, Andrea Dell'Isola.

**Investigation:** Belinda J Lawford, Andrea Dell'Isola.

**Methodology:** Belinda J Lawford, Ali Kiadaliri, Michelle Hall, Andrea Dell'Isola.

**Resources:** Martin Englund, Kim L Bennell, Rana S Hinman.

**Writing – original draft:** Belinda J Lawford, Andrea Dell'Isola.

**Writing – review & editing:** Ali Kiadaliri, Martin Englund, Kim L Bennell, Rana S Hinman, Michelle Hall, Andrea Dell'Isola.

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
