## [Editor Report · Decision Letter 0]

23 Sep 2024

Dear Dr Lawford,

Thank you for submitting your manuscript entitled "Association between change in willingness for joint replacement surgery after an education and exercise program for hip/knee osteoarthritis and probability of receiving actual surgery: A longitudinal cohort study of 55,059 people" for consideration by PLOS Medicine.

Your manuscript has now been evaluated by the PLOS Medicine editorial staff and I am writing to let you know that we would like to send your submission out for external peer review.

Please re-submit your manuscript within two working days, i.e. by Sep 25 2024.

Feel free to email me at atosun@plos.org or us at plosmedicine@plos.org if you have any queries relating to your submission.

Kind regards,

Alexandra Tosun, PhD

Associate Editor

PLOS Medicine

---

## [Decision Letter · Decision Letter 1]

21 Nov 2024

Dear Dr Lawford,

Many thanks for submitting your manuscript "Association between change in willingness for joint replacement surgery after an education and exercise program for hip/knee osteoarthritis and probability of receiving actual surgery: A longitudinal cohort study of 55,059 people" (PMEDICINE-D-24-03148R1) to PLOS Medicine. The paper has been reviewed by subject experts and a statistician; their comments are included below and can also be accessed here: [LINK]

As you will see, the reviewers are supportive of the manuscript and have provided valuable comments and questions, as well as suggestions to improve the exploration of the data. After discussing the paper with the editorial team and an academic editor with relevant expertise, I'm pleased to invite you to revise the paper in response to the reviewers' comments. We plan to send the revised paper to some or all of the original reviewers, and we cannot provide any guarantees at this stage regarding publication.

We ask that you submit your revision by Dec 12 2024. However, if this deadline is not feasible, please contact me by email, and we can discuss a suitable alternative.

Don't hesitate to contact me directly with any questions (atosun@plos.org).

Best regards,

Alexandra

Alexandra Tosun, PhD

Associate Editor

PLOS Medicine

atosun@plos.org

Comments from the reviewers:

Reviewer #1: A very interesting manuscript! This paper examines associations between self-reported surgery willingness before and after an exercise program and receipt of surgery after participation in this program (as a related note, the title was a bit weird. You're not looking at an association between a change in surgery willingness and the /probability/ of receiving surgery, but /whether/ the patient actually receives the surgery). One major strength of the study is the registry-based design that captures a population-level patient experience; choices for patient inclusion/exclusion made sense. Another strength is empirical confirmation of "surgery unwillingness" being associated with being less likely to go through with a surgery and more importantly, the authors' attempt at quantifying the extent to which this occurs. However, I do have some comments regarding the statistical analysis in the manuscript, which as is, was not entirely satisfactory.

Given that the intent wasn't to make the most accurate predictions, but rather to understand any association between receipt of this intervention/status of the intervention and the outcome, why not simply use all identified confounders in the modeling? I didn't understand what the variable selection process was trying to achieve. Even if certain variables such as sex might not have a statistically significant association with receipt of surgery, why not include them in the model so you can say that it was adjusted for? (along those lines, it actually looks like there's an association between sex and patient segmentation in terms of willingness before/after the intervention). As an aside, in the figures, you mentioned that survival curves were estimated based on the adjusted models. Does this specifically mean that for the four curves, these are for the four patient segments, marginalizing over the distribution of all confounders of all patients in each category?

Regarding the causal inference, the use of counterfactuals was certainly interesting from a real world perspective, but causal assumptions did not appear to be mentioned at all beyond assuming that all possible confounders have been accounted for (which is unrealistic here). Is a causal model being asserted, and if so, what is the causal estimand of interest? Does it accurately reflect some yes/no pairs canceling with no/yes pairs? The phrasing regarding "causal framework using counterfactual scenarios" appears in the discussion section, without sufficient justification for use of causal methods or formal evaluation of assumptions in the methodoloy. If you are going to use a causal analysis, it would have been helpful to actually do this formally. As for the actual assumptions themselves, I find it hard to believe that comparing the real-life yes/yes vs. yes/no is truly causal where everything has been accounted for. With that said, despite use of causal methodology and estimation of causal effects, results are phrased in terms of associations/as if you aren't trying to exploit causal inference methodology - this is fine, but just kind of discordant given what the methodology might imply.

An interesting aspect of your data is that there were more no/yes discordant pairs than yes/no discordant pairs, which makes sense. However, it might be of scientific interest to examine what might be associated with each of these types of patients (i.e., what is associated with a patient going from willing to unwilling to undergo surgery?), as well as examine patients who go "the wrong way" in terms of the intervention. Given that the intervention was designed to reduce patient willingness to undergo surgery, why did many of them go from unwilling to willing after the intervention? This might be of potential interest. I found it surprising that the actual results from your regression modeling was not presented at all - I strongly recommend you provide adjusted hazard ratios for your model, as you would be able to answer these types of questions. For instance, a clinician may be interested in whether there might be any potential associations based on age - are older or younger adults more/less likely to undergo surgery, especially considering that the recovery period after the operation may represent a larger burden for older adults? In general, displaying results from regression models would allow readers to understand whether certain other variables might also be associated with differential time to surgery (I'm particularly interested in a model account for all potential confounders assessed, not one based on some likelihood-based variable selection criteria).

As some final minor issues, there is some weirdness in the way results are reported. For instance, it is not clear what "hazard rate" means in line 246 (i.e., do you simply mean the hazard of surgery?).

Reviewer #2: This is a very interesting and relevant research that should be accepted if the authors addressed the following important comments:

-This paper seems to have been written by methodologists for methodologists. It is not so much accessible for the average reader with little quantitative skills, especially in survival analysis and parametric modelling. The reporting is pretty indigestible, even for an applied medical statistician. The grouping should be described in an easier way, with of course a formal description in the methods section and then the use of a reader-friendly wording. For example (but feel free to use any other relevant wording): Never considered surgery vs NO-NO, always planned surgery vs YES-YES, ultimately refused surgery vs YES-NO, or ultimately accepted surgery vs NO-YES . Moreover, the numbers presented in tables should never be repeated in the text, which should only report a digest of the main messages. The figure titles are too technical: Use footnotes to describe the technical elements, including what adjustments have been made, but don't use such hard-core titles that will put off reviewers and readers. In the text wording such as "The hazard rate (i.e., instantaneous risk of surgery for anyone who had not already had surgery at that time point) at 1-year post-intervention among those who responded YES-NO was half of that of 245 those responding YES-YES (hazard ratio 0.5 [95% CI 0.4-0.5]; Table 3; Figure 1)." is really not reader-friendly. Why not writing " the risk of surgery for anyone initially accepting but then refusing surgery at 1-y post-inter was half the risk estimated among patients still willing to undergo surgery…" or something like that?

-One key ignored methodological aspect is the clustering of data by hospital facilities: it is possible that patients attending a surgical place could be influenced by the local clinical/surgical team for or against surgery. So we cannot rule out a cluster effect. The analyses are not accounting for the multilevel nature of the used data. However, this is my experience that conducting multilevel flexible survival analysis is incredibly computer-intensive, and too often with big data, the statistical software (R or Stata) tends to crash. The authors should therefore either consider to model the clustering (Multilevel mixed-effects parametric survival analysis: Estimation, simulation, and application - Michael J. Crowther, 2019 (sagepub.com)) or discuss the potential implications of ignoring this clustering in the limitations section.

-It is unclear whether the willingness to undergo surgery is related to the severity of the symptoms experienced, and therefore the length-of-time with a high level of OA limitations. It could be that patients suffering from severe pain and functional limitations would want surgery to alleviate their symptoms; and vis-versa for those with lower or more recent OA symptoms. The authors are already adjusting for pain levels at baseline and post-intervention, but table 1 is clearly showing that the level of post-intervention pain is the highest in the YES-YES and NO-YES groups. Does it deserve further discussions? It would be dangerous for any care provider to wrongly use this article to reduce arthroplasty provision and related funding based on the overall message of this research and ignoring the specificities of the YES-YES and NO-YES groups. This requires more discussion.

-Minor comments:

*The term "incident joint replacement". What does it mean? This is non-reader friendly. Does it refer to the concept of incidence ratio, but then the concept of person-year is very briefly mentioned. Clarify, but make sure to use some wordings that are simple and accessible for all readers.

*Consider reporting a flow diagram graph rather than appendix 1.

*The number of comorbidities categories is ridiculously large, simplify to 0,1,2 3+

*In the figures, the concept "end of intervention" is difficult to grasp. Consider a more reader-friendly labelling.

Reviewer #3: This is an interesting paper that reports on secondary analyses of population based registry data for people seeking care for OA and people receiving total joint arthroplasty (TJA) in Sweden. The primary question is whether participation in a 3-month education and exercise intervention results in a change in patient willingness to undergo TJA and the relationship of pre-post willingness to subsequent receipt of TJA at 5 and 9 years follow-up. The statistical approaches used to link and analyse the data are sound, addressing potential confounders. The authors acknowledge the limitations of a single yes/no measure of willingness before and after a brief 3-month intervention as causally related to subsequent rates of TJA, yet the simplicity of the approach is compelling. Importantly, the authors found that those who converted from yes i want surgery to no i do not want surgery were less likely to receive a TJA during follow-up... in the short term specifically...compared to the yes-yes group.

My main concerns regarding the study relate to the time lag from the education/exercise intervention and receipt of surgery, and thus the causal relationship, if any. The results are hypothesis generating at best. But, the results are provocative with respect to the potential role of improved OA symptom management to reduce symptoms/improve function in helping stem growing demand for TJA for OA. That said, I have some suggestions to improve the interpretability of the findings to readers and also clarify the extent to which potential confounders have been accounted for.

Abstract: it would be helpful to clarify that all patients presenting for OA care in Sweden are provided the 3-month education and exercise intervention.

Introduction: Lines 74-76- I am not sure what the sentence that begins with "Further, joint replacement..." The sentence appears redundant - please revisit/clarify.

Methods:

Please clarify what changes, if any, were introduced to the education/exercise OA intervention over the 10-year time period of this study. Please also explain why the willingness question was introduced, and justify why a simple yes/no versus Likert scale was used as the measure of willingness.

Confounders:

To what extent was the selection of confounders informed by prior literature on the factors that influence willingness to consider TJA? Prior work has identified social network factors and other social health determinants as key to determining people with OA's perceived candidacy for surgery, perceptions of the risks and benefits of surgery, and perceived OA severity, all of which relate to willingness - please comment more explicity on how these factors were incorporated into analyses.

Was the overall burden of OA considered - i.e., the total number of joints affected?

Results:

Table 1 is very informative - there are expected differences across the groups with respect to confounders controlled for (sex, education attainment, measures of OA symptom severity), yet the results do not help the reader to understand the role of controlling for these differences on the outcomes of interest. For example, what was the effect of controlling for a prior surgeon visit? I think that greater clarity regarding a) which variables were controlled for in which analysis - e.g., footnotes in figures and tables - and mention of the effect of controlling for key confounders on the results, would help readers interpret the results and also plan for future studies to confirm or refute the findings.

Discussion:

Overall, the discussion is well-written and addresses methodological limiations inherent in secondary cohort analyses. However, as noted above, I think the discussion regarding the potential mechanisms by which the intervention resulted in changes in willingness to consider TJA, or not, could be strengthened considerably, building on prior work regardiing factors that influence the willingness construct. From the paper, it appears the primary mechanism considered was change in OA symptoms and improvement in arthritis coping / self-efficacy, which I agree may have played a role. But what about speaking with peers who had undergone TJA previously? What about interactions with surgeons, recommendations for surgery received during the intervention? Greater description of what the intervention entails, and whether surgery is explicitly discussed, and the opportunity for group discussion among patients, would be helpful in this respect.

Page 20, line 322 states that "...symptom progression as well as poor adherence to self-management strategies may have contributed to reconsideration about willingness for surgery." Could you please explain what is meant by "poor adherence" to therapy? how might this be related to change in willingness if not due to progression of symptoms?

Figures and Tables:

Please clarify variables controlled for in the various models in footnotes.

Was the overall burden of OA considered - i.e., the total number of joints affected?

Reviewer #4: Dear Author,

It was a pleasure to review your manuscript.

Well done on pursuing this research topic and using such meticulous methodology and statistical analyses. Congratulations on the amazing results and please consider my comments in the attachments and amend the manuscript accordingly.

Kind regards,

Candice

---

* Please upload any figures associated with your paper as individual TIF or EPS files with 300dpi resolution at resubmission; please read our figure guidelines for more information on our requirements: http://journals.plos.org/plosmedicine/s/figures. While revising your submission, please upload your figure files to the PACE digital diagnostic tool, https://pacev2.apexcovantage.com/. PACE helps ensure that figures meet PLOS requirements. To use PACE, you must first register as a user. Then, login and navigate to the UPLOAD tab, where you will find detailed instructions on how to use the tool. If you encounter any issues or have any questions when using PACE, please email us at PLOSMedicine@plos.org.

* FINANCIAL DISCLOSURES: The funding statement should include: specific grant numbers, initials of authors who received each award, URLs to sponsors’ websites. Also, please state whether any sponsors or funders (other than the named authors) played any role in study design, data collection and analysis, the decision to publish, or preparation of the manuscript. If they had no role in the research, include this sentence: “The funders had no role in study design, data collection and analysis, decision to publish, or preparation of the manuscript.”

* COMPETING INTERESTS: All authors must declare their relevant competing interests per the PLOS policy, which can be seen here: https://journals.plos.org/plosmedicine/s/competing-interests

For authors with ties to industry, please indicate whether any of the interests has a financial stake in the results of the current study.

* DATA AVAILABILITY: The Data Availability Statement (DAS) requires revision. For each data source used in your study:

* ETHICS STATEMENTS: In the ethics statement in the Methods section, please ensure that you have specified (1) whether consent was informed and (2) what type you obtained (for instance, written or verbal, and if verbal, how it was documented and witnessed). If the need for consent was waived by the ethics committee, please include this information. If patients provided informed written consent to have data from their medical records used in research, please include this information.

FIGURES AND TABLES

SUPPLEMENTARY MATERIAL

REFERENCES

* Where website addresses are cited, please include the complete URL and specify the date of access (e.g. [accessed: 12/06/2024]).

STUDY TYPE-SPECIFIC REQUESTS

* Abstract: Please include the study design, population and setting, number of participants, years during which the study took place (enrollment and follow up), length of follow up, and main outcome measures.

* Please ensure that the study is reported according to the RECORD guideline (available from https://www.record-statement.org) and include the completed checklist as Supporting Information. Please add the following statement, or similar, to the Methods: "This study is reported as per the Reporting of Studies Conducted using Observational Routinely-Collected Data (RECORD) guideline (S1 Checklist)." When completing the checklist, please use section and paragraph numbers, rather than page numbers.

* For all observational studies, in the manuscript text, please indicate: (1) the specific hypotheses you intended to test, (2) the analytical methods by which you planned to test them, (3) the analyses you actually performed, and (4) when reported analyses differ from those that were planned, transparent explanations for differences that affect the reliability of the study's results. If a reported analysis was performed based on an interesting but unanticipated pattern in the data, please be clear that the analysis was data driven.

* Please state in the Methods section whether the study had a prospective protocol or analysis plan. If a prospective analysis plan (from your funding proposal, IRB or other ethics committee submission, study protocol, or other planning document written before analyzing the data) was used in designing the study, please include the relevant document(s) with your revised manuscript as a Supporting Information file to be published alongside your study and cite it in the Methods section. A legend for this file should be included at the end of your manuscript. If no such document exists, please make sure that the Methods section transparently describes when analyses were planned, and when/why any data-driven changes to analyses took place. Changes in the analysis, including those made in response to peer review comments, should be identified as such in the Methods section of the paper, with rationale.

---

## [Decision Letter · Decision Letter 2]

13 Feb 2025

Dear Dr. Dell'Isola,

Thank you very much for re-submitting your manuscript "Association between change in willingness for joint replacement surgery after an education and exercise program for hip/knee osteoarthritis and future joint replacement surgery: A longitudinal cohort study of 55,059 people" (PMEDICINE-D-24-03148R2) for review by PLOS Medicine.

Thank you for your detailed response to the editors' and reviewers' comments. I have discussed the paper with my colleagues, and it has also been seen again by three of the original reviewers. The changes made to the paper were mostly satisfactory to the reviewer. As such, we intend to accept the paper for publication, pending your attention to the reviewers' and editors' comments below in a further revision. When submitting your revised paper, please once again include a detailed point-by-point response to the editorial comments.

[LINK]

In revising the manuscript for further consideration here, please ensure you address the specific points made by each reviewer and the editors. In your rebuttal letter you should indicate your response to the reviewers' and editors' comments and the changes you have made in the manuscript. Please submit a clean version of the paper as the main article file. A version with changes marked must also be uploaded as a marked up manuscript file. Please also check the guidelines for revised papers at http://journals.plos.org/plosmedicine/s/revising-your-manuscript for any that apply to your paper.

We ask that you submit your revision within 1 week (Feb 20 2025). However, if this deadline is not feasible, please contact me by email, and we can discuss a suitable alternative.

Please do not hesitate to contact me directly with any questions (atosun@plos.org). If you reply directly to this message, please be sure to 'Reply All' so your message comes directly to my inbox.

We look forward to receiving the revised manuscript. 

Sincerely,

Alexandra Tosun, PhD

Associate Editor 

PLOS Medicine

plosmedicine.org

Comments from Reviewers:

Reviewer #1: The authors have satisfactorily addressed all of my comments. With that said, I now have a minor quibble with respect to a change made due to a comment from the second reviewer. Specifically, hazards are not risks - I would simply call a hazard a hazard - it is the instantaneous rate of failure given that a patient had not yet failed at that time. Neither are hazards unconditional rates, which was my original comment.

The manuscript is sound and interesting, but I would strongly recommend simply using the term "hazard."

Reviewer #3: Thank you for the extensive rewriting and attention to addressing prior questions and concerns. I have no additional issues to consider.

Reviewer #4: Dear Authors,

Thank you very much for considering my comments and recommendations!

I am happy with the way that you had addressed my comments/recommendations and know that this revised manuscript will be a meaningful contribution to our wider research and health professional community.

Kind regards,

Candice

[LINK]

Requests from Editors:

GENERAL

1) Please confirm that your abstract complies with our requirements, including providing all the information relevant to this study type https://journals.plos.org/plosmedicine/s/submission-guidelines#loc-abstract

2) Please ensure that all abbreviations are defined at first use throughout the text.

3) Please review your text for claims of novelty or primacy (e.g. 'for the first time') and remove this language (e.g. line 401). In addition, please check that any use of statistical terms (such as trend or significant) are supported by the data, and if not please remove them.

4) Where data points are discrete, please ensure that they are depicted in the figures as discrete data and not as a continuous line.

5) Statistical reporting: Please separate upper and lower bounds with commas instead of hyphens as the latter can be confused with reporting of negative values. Please revise throughout the manuscript.

6) Citations should be in square brackets, and preceding punctuation. Please revise throughout.

ETHICS

Please include the ethics statement from lines 522-526 in the Methods section of your manuscript.

ABSTRACT

1) Please include basic participant characteristics in the Methods and Findings section (e.g. sex, age, BMI).

2) l.58ff: Please ensure that you include the statistical definitions (e.g. 95% CI, HR, etc.) for each set of brackets. Please revise throughout the Abstract and the main text.

3) In the last sentence of the Abstract Methods and Findings section, please describe the main limitation(s) of the study's methodology.

4) Please ensure that all numbers presented in the abstract are present and identical to numbers presented in the main manuscript text.

5) Please include the important dependent variables that are adjusted for in the analyses.

6) Conclusion: What specific implications (supported by the results) does the study have?

METHODS AND RESULTS

1) Please clarify in the Methods and Results section how joint paint, quality of life, pain self-efficacy and number of comorbidities were measured (i.e. what scale/tool was used, as done below Table 1).

2) Table 1: Please include a unit for 'Age' (years).

3) Please ensure that all tables and figures are (appropriately) referenced in the main text.

4) Figure 2: Please indicate in the figure caption the meaning of lines and shaded areas.

DISCUSSION

Please remove all subheadings including the conclusion subheading.

SUPPLEMENTARY MATERIAL

1) Please ensure that all supplementary files are referenced in the main text.

2) Thank you for providing the RECORD checklist. Please replace the page numbers with paragraph numbers per section (e.g. "Methods, paragraph 1"), since the page numbers of the final published paper may be different from the page numbers in the current manuscript.

General Editorial Requests

---

## [Editor Report · Decision Letter 3]

18 Mar 2025

Dear Dr Dell'Isola, 

On behalf of my colleagues and the Academic Editor, Christelle Nguyen, I am pleased to inform you that we have agreed to publish your manuscript "Association between change in willingness for joint replacement surgery after an education and exercise program for hip/knee osteoarthritis and future joint replacement surgery: A longitudinal cohort study of 55,059 people" (PMEDICINE-D-24-03148R3) in PLOS Medicine.

I appreciate your thorough responses to the reviewers' and editors' comments throughout the editorial process. We look forward to publishing your manuscript, and editorially there are only a few remaining points that should be addressed prior to publication. We will carefully check whether the changes have been made. If you have any questions or concerns regarding these final requests, please feel free to contact me at atosun@plos.org.

Please see below the minor points that we request you respond to (line numbers according to Marked Up Manuscript):

* Title: If you agree, we suggest shortening the title to: Change in willingness for joint replacement surgery after an education and exercise program for hip/knee osteoarthritis: A longitudinal cohort study of 55,059 individuals

* Abstract, l.80, please change to: ‘was then higher at 5-years (HR 1.4 [95% CI: 1.2, 1.7])’. Please ensure that you include the abbreviation for Hazard Ratio in the preceding brackets.

* Table 1: Please define ‘SD’ below the table.

* Data availability: In addition to the two website links, please include email addresses where available. We feel that the links to the general website do not provide a direct link that would allow other researchers to find out who to contact.

Before your manuscript can be formally accepted you will need to complete some formatting changes, which you will receive in a follow up email (including the editorial points above). Please be aware that it may take several days for you to receive this email; during this time no action is required by you. Once you have received these formatting requests, please note that your manuscript will not be scheduled for publication until you have made the required changes.

PRESS

Sincerely, 

Alexandra Tosun, PhD 

Associate Editor 

PLOS Medicine